# Postprandial Blood Glucose and Insulin Response in Healthy Adults When Lentils Replace High-Glycemic Index Food Ingredients in Muffins, Chilies and Soups

**DOI:** 10.3390/nu16162669

**Published:** 2024-08-13

**Authors:** Dita Chamoun, Alison M. Duncan, Patricia K. Lukus, Michael D. Loreto, Frances Pals-Horne, Aileen Hawke, D. Dan Ramdath

**Affiliations:** 1Department of Human Health & Nutritional Sciences, University of Guelph, Guelph, ON N1G 2W1, Canada; dmoravek24@gmail.com (D.C.); pklukus@gmail.com (P.K.L.); michael.loreto02@gmail.com (M.D.L.); franpalshorne@gmail.com (F.P.-H.); 2Guelph Research and Development Centre, Agriculture and Agri-Food Canada, Guelph, ON N1G 5C9, Canada; aileen.hawke@agr.gc.ca

**Keywords:** type 2 diabetes, lentils, glycemic response, insulin, healthy adults

## Abstract

Objectives: This study aimed to assess postprandial blood glucose response (PBGR), relative glycemic response (RGR) and insulin response when 25 g available carbohydrates (AC) is replaced with cooked lentils in the formulation of muffins, chilies and soups. Methods: In randomized, crossover studies, healthy adults consumed foods containing 25 g AC from green lentils, red lentils or a control (wheat muffin, *n* = 24; rice chili, *n* = 24; potato soup, *n* = 20). Blood collected at fasting and at 15, 30, 45, 60, 90 and 120 min was analyzed to derive the incremental area under the response curve (iAUC) for glucose, insulin, RGR and maximum concentration (C_MAX_). Treatment effects were assessed with repeated measures ANOVA. Results: A replacement of 25 g AC with green lentils significantly decreased glucose iAUC compared to chili and soup (*p* < 0.0001), but not muffin (*p* = 0.07) controls, while also eliciting a significantly lower insulin iAUC for all three foods (muffin *p* = 0.03; chili *p* = 0.0002; soup *p* < 0.0001). Red lentil foods significantly decreased glucose iAUC (muffin *p* = 0.02; chili *p* < 0.0001; soup *p* < 0.0001) compared to controls, with a significantly lower insulin iAUC for chili and soup (*p* < 0.0001) but not muffins (*p* = 0.09). The RGR for muffins, chilies and soups was 88, 58 and 61%, respectively, for green lentils, and 84, 48 and 49%, respectively, for red lentils. Conclusions: PBGR, insulin and RGR are decreased when lentils are incorporated into food products, providing credible evidence to promote carbohydrate replacement with lentil-based foods.

## 1. Introduction

Lentils, a member of the pulse family that also includes peas and beans, contain an exceptional nutritional profile that is low in fat and high in protein and fiber, which account for their many associated health benefits [1]. Lentils are high in complex carbohydrates, which attenuate postprandial blood glucose response (PBGR) and have been characterized as a low-glycemic index (GI) food [2]. Epidemiological studies have demonstrated an inverse relationship between the consumption of lentils and other pulses and the risk for type 2 diabetes [3,4]. Additionally, regular pulse consumption alone, or in addition to a low-GI or high-fiber diet, has been associated with improved markers of glycemic control [5]. It is reasonable to promote the increased consumption of lentils as a useful strategy to lower PBGR. However, since it is likely that the food matrix greatly influences starch bioavailability [6], and thus glycemic response, the PBGR-lowering efficacy of lentils within different food matrices needs to be objectively examined.

Acute feeding trials have shown that PBGR following lentil consumption is significantly lower than other starch-rich foods such as pasta or potatoes [7,8]. Lentils also reduced PBGR when consumed in a mixed meal, such as macaroni and tomato sauce [9], or blended in a burrito [10] when compared to the same meal without lentils, but this finding has not been consistent [11]. Additionally, the preparation of food products by replacing starchy ingredients with lentils or pulse ingredients results in meaningful reductions in PBGR and relative glycemic response (RGR) [2,12]. However, many of the existing studies lack a standardized approach for incorporating lentils into food products. For example, studies have standardized the amount of lentils in test meals by energy, volume or available carbohydrates; this may account for the discrepancies in treatment effects with lentils. We have shown that, in a mixed meal, replacing a fixed amount of available carbohydrates (AC) from rice or potato with cooked lentils results in a significant reduction in PBGR in healthy adults [13]. As such, lentils might be regarded as an ideal ingredient for formulating food products that can be used to improve the management and prevention of type 2 diabetes [14], but more studies within different food matrices are needed.

The current consumption of lentils, and other pulses, in North America is insufficient to realize their potential health benefits [15,16]. One approach to address this problem is to promote the increased utilization of lentils as a replacement ingredient in home food preparation and in the manufacturing of value-added food products, since lentil consumption is associated with a reduction in PBGR [2,12,17]. Furthermore, high rice consumption among Asian populations is significantly associated with increased risk of type 2 diabetes [18,19], independent of traditional risk factors [20]. As such, promoting the incorporation of lentils into dietary traditional patterns may improve glycemic control amongst persons who may struggle with dietary advice for disease management that promotes the exclusion of many foods in their traditional diet [21]. Indeed, culturally competent care, including dietary advice that aligns with cultural practices, is important for self-management and treatment adherence [21,22].

Lentils provide an opportunity to manufacture value-added foods and promote a dietary pattern that is culturally inclusive and that confers health benefits [23]. However, the ability of lentils to reduce PBGR when incorporated into complex food matrices has not been properly assessed. In this study, we have used a carbohydrate replacement approach to assess the effect on PBGR in healthy adults, when high-GI food ingredients are replaced by lentils. Specifically, the blood glucose iAUC following the consumption of muffins, rice and chilies made with wheat flour, rice or potato, respectively, were compared with the same foods in which 25 g AC was replaced by either green or red lentils.

## 2. Materials and Methods

### 2.1. Study Design

This study was conducted at the Human Nutraceutical Research Unit (HNRU) at the University of Guelph, was approved by the University of Guelph Research Ethics Board (REB#16AU008) and was registered on ClinicalTrials.gov (NCT02923089). A randomized, crossover study design was used for each of the three treatments (muffin, chili, soup). Participants in each treatment group attended three 3 h study visits (small green lentil, split red lentil, control) separated by three- to seven-day washout periods.

### 2.2. Participant Recruitment and Screening 

Healthy adults aged 18–40 years of age with a body mass index (BMI) of 20–30 kg/m^2^ were recruited from Guelph, ON, Canada and surrounding communities using posters and online and newspaper advertisements. Exclusion criteria included the following: fasting blood glucose > 6.1 mmol/L or >7.8 mmol/L at two hours post 75 g glucose challenge (i.e., oral glucose tolerance test), major medical conditions, medical or surgical events requiring hospitalization within the past 3 months, medication use except stable (3 months) doses of oral contraceptives, blood pressure >140/90 mmHg, use of probiotics, dietary fiber or any other natural health products for glycemic control, consumption of >4 servings of pulses per week, food allergies or non-food life-threatening allergies, pregnancy or breastfeeding, shift work, recent significant weight loss or gain (>4 kg within 3 months), tobacco use, alcohol consumption >14 drinks per week or >4 drinks per sitting and being an elite athlete.

A three-step screening process was implemented in which priority eligibility criteria (i.e., age, self-reported height, self-reported body weight, food allergies, medication use) were assessed in a Screening-1 phone interview. If eligible, participants next attended a Screening-2 visit, during which body measurements (height, body weight, blood pressure) were completed, in addition to a detailed eligibility questionnaire. Eligible participants continued to the final Screening-3 visit for which they arrived to the HNRU after a 10–12 h overnight fast having avoided alcohol, medications, pulses and unusual physical activity for 24 h prior to complete an oral glucose tolerance test. After the consent form was signed, the fasted body weight was measured in duplicate, and the waist circumference was measured by having the participant hold the end (0 cm) of a measuring tape to their navel while the study coordinator ensured the measuring tape remained horizontal and wrapped it around the participant’s waist. A fasted blood sample was collected by finger prick, followed by the consumption of a glucose drink (containing 75 g glucose; #401223, Thermo Scientific NERL Trutol Glucose Tolerance Test Beverage, Thermo Fisher Scientific, Waltham, MA, USA) and a second blood sample at 2 h. Blood was immediately analyzed in duplicate for glucose using a Nova Biomedical StatStrip Glucose Hospital Meter (#53634, Nova Biomedical Canada Ltd., Mississauga, ON, Canada). Eligible participants completed a study-orientation session during which they signed the study consent form. 

### 2.3. Study Treatments

Study treatments included one of three types of muffins, chilies and soups that each contained one of two types of lentils (small green lentils, split red lentils) or a commonly consumed high-GI starch as the control (white wheat, white rice or instant mashed potato) for a total of nine treatments. High-GI foods were specifically chosen as controls as these may be targets for replacement to improve PBGR in the general population. Foods were formulated and produced at the Weston Sensory and Food Research Centre at the University of Manitoba, shipped frozen to the HNRU and stored at −20 °C until ready to use. Treatments were standardized to provide 25 g AC from lentils or the starch-rich control (white wheat, white rice, instant mashed potato) based on glycemic carbohydrates (total starch and free sugars) and proximate analysis (Table 1). On the morning of each study visit, treatments were prepared in the metabolic kitchen of the HNRU by reheating either in the microwave (muffins) or on the stovetop in a saucepan (chilies and soups). Participants consumed all treatments within 10 min with 250 mL of water, except for the soup, with which participants were allowed to drink water ad libitum.

### 2.4. Study Visit Preparation

Before each study visit, participants were instructed to fast for 10–12 h overnight, avoid alcohol, medications, pulses and unusual physical activity for 24 h and to consume the same dinner (of their choice) the evening prior.

### 2.5. Anthropometric Measurements

Body weight was measured in duplicate at each study visit (Acculab^®^ Sartorius Group SVI-200F, Sartorius Stedim Biotech, Aubagne, France) with the participant not wearing shoes and with their pockets emptied. Blood pressure was measured in duplicate after participants were seated for 5–10 min (Omron^®^ Digital Blood Pressure Monitor, HEM-907 XL, Omron Healthcare Inc., Burlington, ON, Canada).

### 2.6. Blood Collection and Analysis

Finger prick blood samples were performed using a contact-activated lancet (Becton Dickinson (BD) Microtainer^®^ Contact-activated Lancet, High Flow, 1.5 mm × 2.0, Ref #366594, BD, Mississauga, ON, Canada) and collected into a microtube (BD Microtainer^®^ MAP Microtube for Automated Processes, 1.0 mg K_2_EDTA; BD, Mississauga, ON, Canada) at fasting and 15, 30, 45, 60, 90 and 120 min after the first bite of the treatment food. Blood samples were immediately analyzed in duplicate for glucose using a Nova Biomedical StatStrip Glucose Hospital Meter (Ref #53634, Nova Biomedical Canada Ltd., Mississauga, ON, Canada), which had an intra-assay variation of 2.0% and an inter-assay variation of 9.7% and 6.0% for low and high controls, respectively. Remaining blood was centrifuged (Beckman Coulter Allegra X-22R, Beckman Coulter, Mississauga, ON, Canada; or Thermo Scientific Sorvall ST8R centrifuge, Thermo Fisher Scientific, Waltham, MA, USA) within 15 min at 3000× *g* rpm for 10 min at 4 °C. Plasma was aliquoted into 1.2 mL cryovials (Corning^®^ 1.2 mL Internal Threaded Polypropylene Cryogenic Vial, Self-standing with Conical Bottom, Ref #430487, Corning Inc., Corning, NY, USA) and stored at −80 °C. Plasma samples were analyzed in duplicate for insulin by ELISA (Mercodia, Insulin ELISA, Ref #10-1113-10, Mercodia AB, Uppsala, Sweden), which had an intra-assay variation of 3.6% and an inter-assay variation of 11.5% and 8.6% for low and high controls, respectively.

### 2.7. Data and Statistical Analysis

A total of 20–24 participants were included in each of the three studies (muffin, chili, soup) to account for possible attrition while enabling the detection of a minimum 20% reduction in blood glucose iAUC at a 5% significance with 80% power. Postprandial blood glucose and plasma insulin response curves for each treatment were calculated as the 2 h incremental area under the curve (iAUC) using the trapezoid rule [24]. Maximum blood glucose concentration (C_MAX_) and time of maximum concentration for each response curve were determined by inspection in Microsoft Excel^®^ (Microsoft Office 2016, Redmond, WA, USA) spreadsheets. Relative glycemic response (RGR) and relative insulinemic response were calculated by dividing the iAUC of lentil treatments by the iAUC of the controls (wheat muffin, rice chili, potato soup) and multiplying the ratio by 100 [24]. Data were examined for normality using stem leaf diagrams and box plots, and it was determined that insulin data required natural log transformation. Insulin data are therefore presented as geometric means and 95% confidence intervals. All variables were assessed for treatment effects within each study (muffin, chili, soup) using repeated measures ANOVA and post hoc Tukey’s tests for multiple comparisons. All statistical analyses were performed using the Statistical Analysis System (version 9.4, Cary, NC, USA) with *p* < 0.05 considered significant.

## 3. Results

### 3.1. Participant Flow and Characteristics

Twenty-four participants were randomized to each of the muffin and chili treatments and twenty to the soup treatment, as shown in Figure 1. Participant baseline characteristics are summarized in Table 2.

### 3.2. Postprandial Blood Glucose and Insulin Response

The postprandial blood glucose response followed a similar pattern for all treatments: increasing from baseline, most frequently peaking at 30 min and returning to baseline at 120 min (Figure 2A–C). The green lentil muffin significantly (*p* < 0.05) decreased blood glucose at 15 and 30 min compared to the wheat muffin, while the red lentil muffin significantly (*p* < 0.05) decreased blood glucose at 45 min compared to the wheat muffin (Figure 2A). Despite these individual timepoint effects, only the red lentil muffin significantly (*p* = 0.02) decreased blood glucose iAUC compared to the wheat muffin, with no significant differences in blood glucose C_MAX_ (Table 3). In the chili study, both green and red lentil chilies significantly (*p* < 0.05) decreased blood glucose at 15, 30 and 45 min (Figure 2B), and significantly (*p* < 0.0001) decreased blood glucose iAUC and C_MAX_ (Table 3) compared to the rice chili. In the soup study, both green and red lentil treatments significantly (*p* < 0.05) decreased blood glucose at 15, 30 and 45 min, which was significantly (*p* < 0.05) higher at 90 and 120 min (Figure 2C), and then significantly (*p* < 0.0001) decreased blood glucose iAUC and C_MAX_ (Table 3) compared to the potato soup. The RGRs elicited by green lentil muffin, chili and soup compared to the controls were 88, 58 and 61%, respectively; this translates to a reduction of 12, 42 and 39% in blood glucose iAUC, respectively. The RGRs of the red lentil muffin, chili and soup compared to the controls were 84, 48 and 49%, respectively; this translates to a 16, 52 and 51% reduction in blood glucose iAUC, respectively.

The postprandial plasma insulin response followed a similar time course pattern to that of blood glucose for all treatments, and there were significant differences by treatment and time (Figure 2D–F). In the muffin study, only green lentil muffins significantly (*p* < 0.05) decreased plasma insulin at 15 min (Figure 2D) as well as plasma insulin iAUC (*p* < 0.05) (Table 3) compared to the wheat muffin, with no significant differences in C_MAX_ (Table 3). In the chili study, time point effects were more varied in terms of plasma insulin as when compared to the rice chili control as follows: significantly (*p* < 0.05) decreased at 15 and 30 min by the green and red lentil chilies; significantly (*p* < 0.05) decreased at 45 min by only the red lentil chili; significantly (*p* < 0.05) increased at 90 min by the green lentil chili; and significantly (*p* < 0.05) increased at 120 min by the red lentil chili (Figure 2E). Overall, both the green and red lentil chilies significantly decreased plasma insulin iAUC and C_MAX_ (iAUC, *p* = 0.0002, *p* < 0.0001, respectively; C_MAX_ *p* < 0.0001 for both) compared to the rice chili (Table 3). In the soup study, both green and red lentil soups significantly decreased plasma insulin at 15, 30, 45 and 60 min, but this was then significantly increased at 90 and 120 min, as compared to the potato soup (*p* < 0.05 for all) (Figure 2F). Both green and red lentil soups significantly (*p* < 0.0001) decreased plasma insulin iAUC and C_MAX_ compared to the potato soup (Table 3). The relative insulinemic response elicited by the green lentil muffin, chili and soup compared to the controls was 86, 75 and 46%, respectively; this translates to a reduction of 14, 25 and 54% in plasma insulin iAUC, respectively. The relative insulinemic response of the red lentil muffin, chili and soup compared to the controls was 92, 63 and 45%, respectively, compared to the controls; this translates to an 8, 37 and 55% reduction in plasma insulin iAUC, respectively.

## 4. Discussion

In this high quality, adequately powered human clinical trial, we assessed the effects of partially substituting commonly consumed, high-GI carbohydrate foods with lentils in complex food matrices including muffins, chilies and soups. Importantly, the dose of lentils (25 g AC) used in this study is an amount that can be reasonably achieved in a single meal. The results also show that, while the attenuation of blood glucose iAUC may be influenced by lentil variety for the muffin treatments, this was not the case for chilies or soups, for which both lentil varieties significantly lowered blood glucose iAUC and C_MAX_. The difference between lentil varieties seen only for the muffin treatments may have been due to changes in starch structure during the different cooking processes, or was perhaps due to the interaction between ingredients, given that wheat flour and added sugar were present in all treatments (i.e., this was a partial replacement with lentils). The identification of putative mechanisms for differences in lentil varieties and PBGR was beyond the scope of this study. Plasma insulin response was always the same or lower than the control for both lentil treatments, which is highly desirable in maintaining normal glycemic control [25,26]. Thus, substituting lentils for other carbohydrate sources can attenuate PBGR in a variety of food matrices, which provides consumers with a variety of culturally relevant dietary options that can contribute to the maintenance of normal blood glucose and possibly reduce the risk of type 2 diabetes.

While other studies incorporated pulses into mixed-meal treatments, none used pulses as an ingredient in a complex, prepared food product, standardized by AC, nor replaced a portion of a commonly consumed carbohydrate. For example, Mollard et al. used relatively simple meal combinations of pulses, pasta and tomato sauce [9,11] or pulses and tomato sauce [27] in their studies on the PBGR-lowering potential of pulses. In one such study, treatments that contained 40 g AC from pulses (total AC ~100 g) did not significantly differ in their effects on blood glucose iAUC when compared to a macaroni and cheese control [11]. However, interpretation is challenged by the finding that cheese, due to its fat content, has been shown to decrease PBGR when consumed with a high-GI food [28]. In another study, pulse treatments were fed ad libitum, resulting in a significant reduction in blood glucose iAUC for all pulse treatments compared to a pasta and tomato sauce control [9]. Mollard et al. also used treatments standardized to 300 kcal, which resulted in a significant decrease in mean blood glucose with pulse treatments compared to a white bread and tomato sauce control [27]. Collectively, these studies demonstrate a PBGR-lowering effect of pulse treatments, and, although pulses were not incorporated as an ingredient into food products, the general findings are consistent with those of our study. Also different in the current study is the use of lentils as a food ingredient in a manner that standardized the amount of AC and replaced high-GI ingredients, which could guide meal replacements and the manufacturing of value-added foods that lower PBGR.

The PBGR-lowering effect of pulses was examined within the context of a complete meal in three earlier studies [7,29,30]. Jenkins et al. compared a breakfast meal of lentils, butter and tomatoes to that of a carbohydrate-matched wholemeal bread, cottage cheese and tomatoes meal, and found that the lentil meal decreased the blood glucose area under the curve by 71% compared to the wholemeal bread meal [29]. Bornet et al. also used minimal additions of cheese and butter with lentils, beans, spaghetti, rice, potato or white bread, in addition to standardizing treatments to 50 g AC from the carbohydrate source, which is similar to the method of the current study [7]. 

The glycemic index of lentils and beans was significantly lower compared to bread and potato and was significantly lower after beans (but not lentils) compared to rice [7]. This latter study, as well as our previous results [13], provide more evidence that pulses can reduce PBGR relative to several other carbohydrate sources, even when supplying the same amount of AC. Coulston et al. also compared different lentils with other carbohydrate sources (rice, potato and spaghetti) in combination with white bread, turkey, margarine, oil and lettuce, while keeping the macronutrient content equal between treatments [30]. The plasma glucose responses for lentils, rice and spaghetti were similar, but were higher for potato [30]. This approach is similar to the current study in having consistent ingredients while only varying the carbohydrate source. However, Coulston et al. standardized macronutrients (including total carbohydrates) across treatments [30], so it is possible that the AC varied. Finally, a 2014 study by Anguah et al. assessed whole or blended lentils in a burrito compared to a lentil-free burrito [10]. Lentil-containing treatments provided ≥½ cup lentils, creating a realistic and culturally relevant meal. Only the blended lentils lowered PBGR compared to the control [10], suggesting an effect of processing on this response. A similar finding was reported by Ramdath et al. [2] who showed that the GI of whole cooked lentils changed from 25 to 66 when lentils were processed by spray-drying.

While the present study aimed to fill several of the gaps in the current literature and create a more cohesive understanding of the PBGR to pulses, it is limited by a lack of blinding, which often occurs in whole food studies, although placebo effects on PBGR using whole foods have not been previously shown. Furthermore, only healthy participants were included, but other populations should be tested in future studies (e.g., postmenopausal women, individuals using medications and those with metabolic abnormalities). Additionally, information about the mechanism by which lentils lower PBGR was not examined. The study has many strengths, including the use of a randomized, crossover design and the realistic treatment design that considered the feasibility of consumption, palatability and cultural relevance. The study used carbohydrate-rich foods in a format that is typically consumed (e.g., wheat in a muffin) and included an amount of lentils that is feasible to consume. The treatments included two market-class lentil varieties, one with a hull and one without. Since lentils require little preparation, the treatments used recipes that are very simple for consumers to make themselves. Lastly, these data show that a practical approach of replacing one carbohydrate with another can have measurable health benefits; this allows policy-makers to endorse evidence-based dietary approaches to type 2 diabetes management and prevention. Additionally, healthcare providers can provide easily adopted lifestyle strategies that move away from an elimination philosophy (e.g., eliminate all refined carbohydrates) and can be incorporated into a variety of traditional diets.

## 5. Conclusions

In conclusion, we found that the partial or full replacement of high-GI, carbohydrate-rich foods with lentils can reduce postprandial blood glucose iAUC and C_MAX_ as well as plasma insulin in healthy adults. While this result may depend on the type of lentil and the amount substituted, lentils appear to be beneficial in attenuating postprandial glycemic response. Dietary changes will be critical in combating global chronic disease issues and are more likely to be accepted if they are convenient and do not sacrifice enjoyment. These data support the creation of more foods that will meet these criteria while reducing disease risk.

## Figures and Tables

**Figure 1 nutrients-16-02669-f001:**
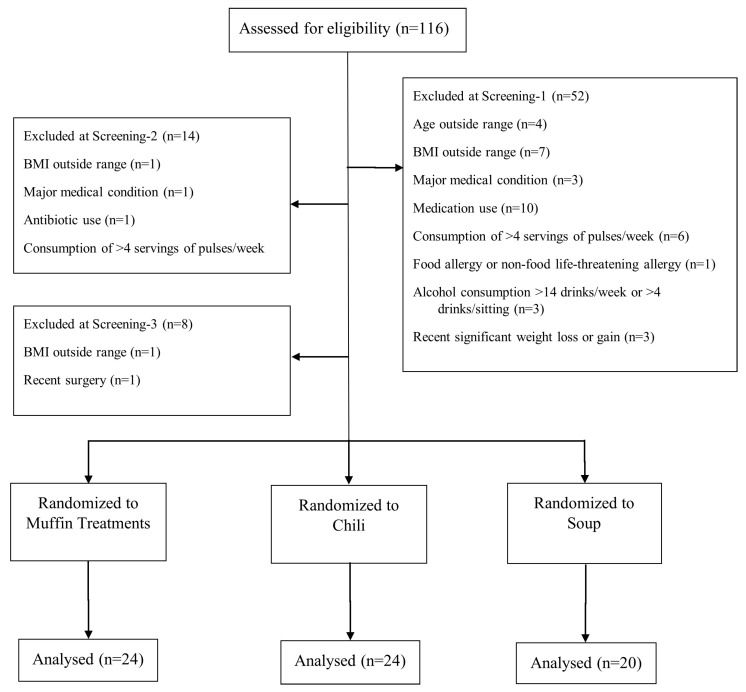
Participant flow diagram. A total of 42 participants were randomized: 4 completed all three studies; 18 completed two studies and 20 completed one study. BMI = body mass index.

**Figure 2 nutrients-16-02669-f002:**
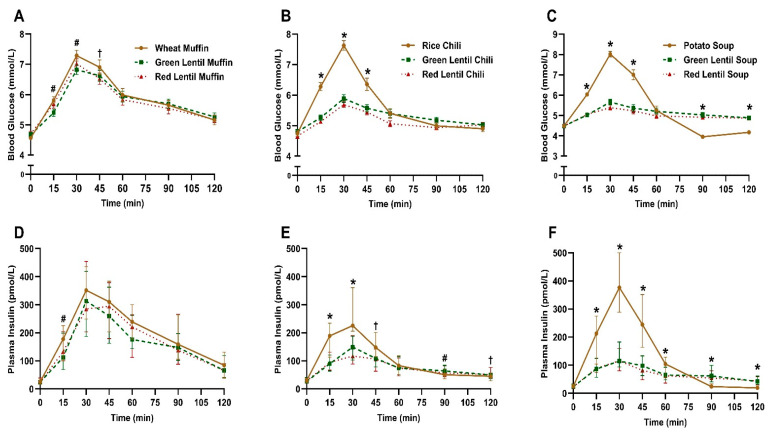
Postprandial blood glucose (top panel (**A**–**C**)) and plasma insulin (bottom panel (**D**–**F**)) response curves of healthy adult participants after muffin (*n* = 24), chili (*n* = 24) and soup (*n* = 20) treatments. Blood glucose values are means ± SE. Plasma insulin values are medians and 95% confidence intervals. * Both lentil treatments were significantly different from control (*p* < 0.05); # red lentil treatment was significantly different from control (*p* < 0.05); † green lentil treatment was significantly different from control (*p* < 0.05).

**Table 1 nutrients-16-02669-t001:** Nutritional composition of study treatments.

	Muffin Treatments			Chili Treatments			Soup Treatments		
	Wheat Control	Green Lentil	Red Lentil	Rice Control	Green Lentil	Red Lentil	Potato Control	Green Lentil	Red Lentil
Serving * (g)	128	204	178	237	305	283	471	361	339
Energy ^†^ (kcal)	384	505	458	194	314	287	190	276	245
Protein ^‡^ (g)	9.5	20.0	16.8	4.8	17.0	14.7	5.6	15.7	13.3
Fat ^§^ (g)	9.9	10.4	10.3	2.2	3.3	3.0	1.3	1.7	1.6
Carbohydrates ^║^ (g)	58.3	71.5	65.2	35.9	46.8	43.5	36.7	43.1	38.3
Dietary Fiber ^¶^ (g)	4.2	15.1	9.8	5.0	16.9	13.1	5.8	13.9	9.7
Available Carbohydrates ^#^ (g)	44.8	46.7	45.4	27.3	26.1	26.6	26.9	26.3	25.3

* Moisture determined via AOAC 930.15 (Central Testing Laboratory, Winnipeg, MB, Canada, from 3 combined batches of prepared food product) and mass loss analysis following freeze-drying (Guelph Research and Development Centre; *n* = 2). ^†^ Bomb calorimeter ANSI/ASTM D2015-77 (University of Arkansas); *n* = 6. ^‡^ Protein (University of Arkansas and Central Testing Laboratory) using AOAC 990.03/990.03; *n* = 6. ^§^ Fat analyzed at the Central Testing Laboratory, 920.39C/AOCS Am5-04; *n* = 3. ^║^ Carbohydrates calculated by difference. ^¶^ Dietary fiber analyzed using AOAC 991.43.3, GOS extracted during the DF procedure were quantified using HPAEC-PAD [23]; *n* = 6. ^#^ Available carbohydrates = [total starch − resistant starch] + free sugars; Total starch (AOAC 996.11 via Megazyme kit method b and resistant starch (AOAC 2002.02) as outlined by kit manufacturer; Megazyme International Ireland Ltd., Bray, Ireland. Free sugars were measured inhouse using established methods [23]; *n* = 3.

**Table 2 nutrients-16-02669-t002:** Participant baseline characteristics for muffin, chili and soup treatment groups ^1^.

	Muffin (*n* = 24)	Chili (*n* = 24)	Soup (*n* = 20)
Age (years)	26.9 ± 1.3	26.6 ± 1.1	23.1 ± 0.7
Sex (*n* male/female)	9/15	10/14	9/11
Body Weight (kg)	69.5 ± 2.2	71.3 ± 2.2	69.3 ± 2.0
BMI ^2^ (kg/m^2^)	24.9 ± 0.4	24.2 ± 0.4	23.8 ± 0.5
Waist Circumference (cm)	80.4 ± 1.6	81.7 ± 1.5	80.2 ± 1.4
Systolic Blood Pressure (mmHg)	117 ± 3	117 ± 3	116 ± 3
Diastolic Blood Pressure (mmHg)	70 ± 2	69 ± 2	68 ± 2

^1^ Values are mean ± SE unless otherwise indicated. ^2^ BMI = body mass index.

**Table 3 nutrients-16-02669-t003:** Postprandial blood glucose and plasma insulin response in healthy adults (*n* = 24) following the consumption of control versus lentil-containing food products *.

	Muffin Study			Chili Study			Soup Study		
	Wheat Control	Green Lentil	Red Lentil	Rice Control	Green Lentil	Red Lentil	Potato Control	Green Lentil	Red Lentil
Postprandial blood glucose ^†^ mmol/L.min	169.0 ± 12.8 ^a^	142.6 ± 14.9 ^a,b^	136.6 ± 13.2 ^b^	122.4 ± 11.0 ^a^	65.4 ± 7.4 ^b^	61.3 ± 10.0 ^b^	130.3 ± 9.8 ^a^	80.1 ± 9.3 ^b^	64.9 ± 9.1 ^b^
C_MAX_, mmol/L	5.1 ± 0.1 ^a^	5.1 ± 0.1 ^a^	5.1 ± 0.1 ^a^	7.7 ± 0.2 ^a^	6.0 ± 0.1 ^b^	5.7 ± 0.1 ^b^	8.0 ± 0.1 ^a^	5.8 ± 0.1 ^b^	5.6 ± 0.1 ^b^
Postprandial plasma insulin ^‡^, nmol/L.min	18.2 (15.1, 21.9) ^a^	15.1 (12.3, 18.5) ^b^	15.7 (13.0, 18.9) ^a,b^	8.4 (7.1, 9.9) ^a^	5.8 (4.7, 7.1) ^b^	4.9 (3.7, 6.4) ^b^	11.2 (9.1, 13.8) ^a^	4.4 (3.1, 6.2) ^b^	4.7 (3.8, 5.9) ^b^
C_MAX_, nmol/L	0.33	0.28	0.30	0.24	0.13	0.12	0.32	0.11	0.11
	(0.27, 0.41) ^a^	(0.23, 0.35) ^a^	(0.25, 0.35) ^a^	(0.21, 0.28) ^a^	(0.11, 0.16) ^b^	(0.10, 0.15) ^b^	(0.27, 0.39) ^a^	(0.09, 0.13) ^b^	(0.09, 0.13) ^b^

* For each food product, row values with different superscript letters are significantly different (*p* < 0.05). ^†^ Values are means ± SE. ^‡^ Values are geometric means (95% confidence intervals).

## Data Availability

The datasets generated during and/or analyzed during the current study are not publicly available due to Crown Copyright, but are available from the corresponding author upon reasonable request.

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
