# Peer review of "Postprandial Blood Glucose and Insulin Response in Healthy Adults When Lentils Replace High-Glycemic Index Food Ingredients in Muffins, Chilies and Soups"

_nutrients, 2024, doi:10.3390/nu16162669_

Round 1

Reviewer 1 Report

Comments and Suggestions for Authors

Chamoun et al report on randomized crossover trials examining the effect of replacing high glycemic index carbohydrates with lentils in a variety of foot matrices on postprandial glycemic and insulin responses. This work helps address a gap in availability of credible evidence regarding the impact of food matrix responses on the ability of lentils to impact postprandial responses. The study is well-conceptualized and the results properly analyzed and presented.  

Introduction: 

The rationale for this study is strong; however, the following additions are recommended to strengthen the rationale even more: 

35 - The authors discuss habitual pulse consumption (line 35) “Additionally, habitual pulse consumption alone, or in addition to a low-GI or high-fibre diet, has been associated with improved markers of glycemic control [5].” However, a habitual consumption study specific to lentils would strengthen this statement: Chamberlin et al. 2024 Jan 31;16(3):419. doi: 10.3390/nu16030419. 

37-38 – The impact of food matrices is central to this paper. A sentence to provide foundational information on food matrices is recommended. Why do we need to know more about this? 

44-45 – Recommend providing an example to illustrate point regarding lack of standardized approaches. 

Methods: 

113 – Please provide a one sentence rationale for selection of white wheat, white rice, and instant mashed potatoes as experimental comparators. Were you specifically seeking high GI foods? 

Study Treatments: 

Why was the available carbohydrate in the chili and soup so low? Mention was made of a standardized approach being needed. Identify what you standardized across these three treatments. Was it just 25 g carbohydrate from lentil sources? This was quite different proportion of carbohydrate across conditions. 

Results: 

Figure 2 is very difficult to read – a combination of poor resolution and small font size. Advise improving quality. 

Table 3 – Should units for iAUC be mmol/L.120 min? (mmol/L.min would be the average over the 120 period rather than the area). I recognize that methods for presenting iAUC units are not consistent across research studies and the method you use is common, I just don’t understand it. Please explain your rationale for me to better understand if you choose to keep the units as they are. Same question for insulin iAUC. 

Discussion: 

Recommend rewriting the first sentence of each paragraph to highlight the key point added to the research by the current study. This would help to emphasize the impact of the current study, which is somewhat buried in the current discussion. Overall, there is a lot of information presented in the discussion without enough context to emphasize what the current study is adding and why it is important.  

One concern regarding the study design that needs to be addressed is that the lentil and control treatments do have comparable available carbohydrate, but all lentil meals have substantially greater total energy. Similarly, another point that should be more fully addressed in the discussion is why the muffin condition differed from the soup and chili condition with respect to impact of the lentils. Does this have to do with replacing a smaller proportion of available carbohydrate? Does this have to do with the greater overall energy content of the muffins?  

The authors utilized two different lentil types in this study and three different meal matrices adding additional information regarding the health impact of lentils on postprandial glycemic measures. However, discussion regarding the compositional differences between lentil types and how/if this may account for the differential impacts on study measures is lacking. The same comment holds for differences in lentil preparation/cooking methodology between the different meal types. As these factors were a major part of the study design, these topics need to be adequately addressed in the discussion. 

250-251 – Awkward sentence.  

268-275 – What does this have to do with the current study? The relevance of this information to the current study is not clear. 

Author Response

Reviewer#1

Chamoun et al report on randomized crossover trials examining the effect of replacing high glycemic index carbohydrates with lentils in a variety of foot matrices on postprandial glycemic and insulin responses. This work helps address a gap in availability of credible evidence regarding the impact of food matrix responses on the ability of lentils to impact postprandial responses. The study is well-conceptualized and the results properly analyzed and presented.  

 Introduction: 

The rationale for this study is strong; however, the following additions are recommended to strengthen the rationale even more: 

Response:

Thank you for your comment and for your time strengthening our manuscript.

35 - The authors discuss habitual pulse consumption (line 35) “Additionally, habitual pulse consumption alone, or in addition to a low-GI or high-fibre diet, has been associated with improved markers of glycemic control [5].” However, a habitual consumption study specific to lentils would strengthen this statement: Chamberlin et al. 2024 Jan 31;16(3):419. doi: 10.3390/nu16030419. 

Response:

Thank you for this comment and we agree that the word “habitual” could be misleading to the reader.  We have therefore replaced this word.  Further since the reference cited is a meta-analysis that examined 39 studies, the reader will be provided with access to substantial evidence. 

37-38 – The impact of food matrices is central to this paper. A sentence to provide foundational information on food matrices is recommended. Why do we need to know more about this? 

Response:

Thank you for this comment and we have added examples to clarify this point on page 2, line 48:

“For example, studies have standardized the amount of lentils in test meals by energy, volume, or available carbohydrates”

Methods: 

113 – Please provide a one sentence rationale for selection of white wheat, white rice, and instant mashed potatoes as experimental comparators. Were you specifically seeking high GI foods? 

Response:

Thank you for pointing this out.  We have added “high-GI” in the description of the foods (page 3, line 116) as well as a rationale for choosing high-GI controls (page 3, line 117):

“High-GI foods were specifically chosen as controls as these may be targets for replacement to improve PBGR in the general population.”

Study Treatments: 

Why was the available carbohydrate in the chili and soup so low? Mention was made of a standardized approach being needed. Identify what you standardized across these three treatments. Was it just 25 g carbohydrate from lentil sources? This was quite different proportion of carbohydrate across conditions. 

Response:

The treatments were standardized to contain 25 g of available carbohydrates from lentils. The muffins had higher available carbohydrate content due to non-lentil ingredients like sugar, which was needed for taste, as well as some wheat flour, without which a desirable texture would not have been achievable. So, while the values for available carbohydrates are higher in the muffins, lentils still provide the standard 25 g of available carbohydrates. Additionally, as treatments were not compared across food form (i.e. muffin vs chili), the difference in available carbohydrates was considered an issue and, importantly, the available carbohydrates was similar between lentil and control treatments within each food form.

Results: 

Figure 2 is very difficult to read – a combination of poor resolution and small font size. Advise improving quality. 

Response:

Thank you for this feedback. The figure has been redone at a higher resolution and font size.

Table 3 – Should units for iAUC be mmol/L.120 min? (mmol/L.min would be the average over the 120 period rather than the area). I recognize that methods for presenting iAUC units are not consistent across research studies and the method you use is common, I just don’t understand it. Please explain your rationale for me to better understand if you choose to keep the units as they are. Same question for insulin iAUC. 

 Response:

We followed Brouns et al (https://pubmed.ncbi.nlm.nih.gov/19079901/) for methodological guidance. In depth formulas can be found on page 162 of the article. In short, since we used incremental area (AUC), in a graph with mmol/L on the y-axis and time (min) on the x-axis, an increment would be calculated using those dimensions as the length and width of a “rectangle” and adjusting these “rectangles” made by each increment in various ways to approximate the shape of the blood glucose curve. Thus, the unit on the x axis is not 120 min, but min, as the time intervals vary.

Discussion: 

Recommend rewriting the first sentence of each paragraph to highlight the key point added to the research by the current study. This would help to emphasize the impact of the current study, which is somewhat buried in the current discussion. Overall, there is a lot of information presented in the discussion without enough context to emphasize what the current study is adding and why it is important.  

Response:

Thank you for this comment. We have added some introductory sentences to paragraphs in the discussion section. Overall, the point in lines 255-300 is that PBGR following pulse consumption has been studied in a variety of ways and interpretation has been made difficult due to the heterogeneity among studies, but also the lack of standardization of treatments within a study. The goal of our study was to follow best practices in studying PBGR, and use a meal design that can reflect a realistic meal. If an improvement in PBGR can be found with simple and palatable substitutions, this could be an area of focus for innovation in food products with health benefits.

One concern regarding the study design that needs to be addressed is that the lentil and control treatments do have comparable available carbohydrate, but all lentil meals have substantially greater total energy. Similarly, another point that should be more fully addressed in the discussion is why the muffin condition differed from the soup and chili condition with respect to impact of the lentils. Does this have to do with replacing a smaller proportion of available carbohydrate? Does this have to do with the greater overall energy content of the muffins?  

 Response:

Thank you for this comment and highlighting this important discussion. The difference in energy content of the treatments is a challenge for many whole food studies. Since we wanted to replace a certain amount of available carbohydrates from a high-GI starch and keep everything else the same, we had to accept energy differences; this is unlike many studies that focused on energy content whilst ignoring differences in available carbohydrates. To increase the amount of energy in the control to that of the lentil treatments would require significant additions of other ingredients that would have brought into question the true effect of the lentils. Importantly, that additional energy is from the lentils and as we are interested in the glycemic impacts of the whole lentil, we must accept all changes to nutritional composition brought about by the addition of lentils.

The difference in glycemic impact between lentil varieties seen only within the muffin treatments is indeed interesting and we have added some proposed explanations on page 9, line 249:

“The difference between lentil varieties seen only for the muffin treatments may have been due to the impact on starches from cooking the lentils before baking them as well;  or perhaps some sort of interaction between ingredients such as wheat flour and added sugars that were present in all treatments (i.e., this was a partial replacement with lentils)”

The authors utilized two different lentil types in this study and three different meal matrices adding additional information regarding the health impact of lentils on postprandial glycemic measures. However, discussion regarding the compositional differences between lentil types and how/if this may account for the differential impacts on study measures is lacking. The same comment holds for differences in lentil preparation/cooking methodology between the different meal types. As these factors were a major part of the study design, these topics need to be adequately addressed in the discussion. 

Response:

Thank you for this important comment, which we have addressed in our revision to this section of the manuscript. It is not clear what accounts for the different effect of lentil varieties or cooking methods on PBGR. This may be due to slight differences in starch composition due to growing conditions or cooking. Importantly, our study did not seek to determine these effects; we used standardized formulation and cooking methods to produce test foods made from common market varieties of lentils, then assessed the comparative effects of each type of food made with or without lentils.

250-251 – Awkward sentence.  

Response:

Thank you. The sentence has been revised (258-260)

268-275 – What does this have to do with the current study? The relevance of this information to the current study is not clear. 

Response:

Thank you. Inclusion of these studies speaks to your previous comment above “This would help to emphasize the impact of the current study, Our goal in these paragraphs of the discussion is to highlight other studies that have used pulses in a mixed meal context, and discuss similarities, differences and knowledge gaps that were filled by our study. This, then, hopefully illustrating where our study fits in among the current research.

Reviewer 2 Report

Comments and Suggestions for Authors

The paper “Postprandial blood glucose and insulin response in healthy adults when lentils replace high glycemic index food ingredients in muffins, chilies and soups” contributes to the growth of literature for research on the area the food for diabetics

However, the following items should be revised:

Introduction

The authors did not describe whether there are any contraindications to consumption of lentils, risk groups.

The authors did not formulate a research aim,  hypothesis.

Methods

Line 112

The authors did not specify:

-          what amount of tested additives was added to muffins, chilies and soups

-          where the products came from

-          composition and method of preparation of products, especially lentils

Results

The Figures 2 are not very visible.

 “P “ - it should be Italic (p), similar to others,

Line 312 “partial replacement” - How partial

Line 316 – 317

The sentence “Dietary changes will be critical in combating global chronic disease issues and are more likely to be accepted if they are convenient and do not sacrifice enjoyment.”

Did the authors investigated the "convenient  and enjoyment" of these products? Did the authors subject the samples to sensory analysis?

How does  the type of product, e.g. muffin or soup, matter?

What are the strengths and limitations of this research?

Author Response

Reviewer#2

The paper “Postprandial blood glucose and insulin response in healthy adults when lentils replace high glycemic index food ingredients in muffins, chilies and soups” contributes to the growth of literature for research on the area the food for diabetics

However, the following items should be revised:

Introduction

The authors did not describe whether there are any contraindications to consumption of lentils, risk groups.

Response:

Thank you. Contraindications to lentil in most cases is due to rare allergy to lentil protein. In studies, such as ours there is sometimes increased flatulence and abdominal discomfort, but rarely are there contraindications.

The authors did not formulate a research aim,  hypothesis.

Response:

Our study aim can be found on page 2, line 72:

“Specifically, the blood glucose iAUC following consumption of muffins, rice, and chilies made with wheat flour, rice or potato, respectively, were compared with the same foods in which 25 g AC was replaced by either green or red lentils.”

Methods

Line 112

The authors did not specify:

-  what amount of tested additives was added to muffins, chilies and soups

-  where the products came from

-  composition and method of preparation of products, especially lentils

Response:

Thank you. It was not our intention to examine the effect of additives on PBGR, given that these are used in small amounts and more importantly they were standardized in the production of each of the muffin or chili or soup prepared.

Line 118-120: The products were prepared Foods were formulated and produced at the Weston Sensory and Food Research Centre at the University of Manitoba, shipped frozen to the HNRU and stored at -20C until ready to use.  

In order to prepare a succinct communication, we did not think it was necessary to report on the ingredients and preparation of the foods, as these are widely available. Further, we thought this was unnecessary because each group of foods (ie, muffin, chili, soup) contained the same ingredients, except for replacement of the high GI ingredient with whole lentil (see Study Treatments). We did not seek to compare eg. chili with muffin or soup with chili, etc. because of differences in ingredient and matrix.  

Results

The Figures 2 are not very visible.

 Response:

Thank you for this feedback. The figure has been redone at a higher resolution and font

 “P “ - it should be Italic (p), similar to others,

 Response:

Thank you for this comment, we have made these changes.

Line 312 “partial replacement” - How partial

 Response:

Thank you for pointing this out. This has been changed to “partial or full replacement”, as lentils partially replaced wheat flour in the muffins, and entirely replaced rice and potato in the chili and soup, respectively.

Line 316 – 317

The sentence “Dietary changes will be critical in combating global chronic disease issues and are more likely to be accepted if they are convenient and do not sacrifice enjoyment.” Did the authors investigated the "convenient  and enjoyment" of these products? Did the authors subject the samples to sensory analysis? How does the type of product, e.g. muffin or soup, matter?

Response:

Thank you for this comment. Participants completed a brief sensory evaluation (taste, texture, appearance etc.) of the treatments but these data were not thorough and therefore not appropriate to include in the current manuscript. In addition, as this statement is in the final paragraph our discussion section, it was meant to imply that convenient and enjoyable foods will be necessary. Our study provides evidence of PBGR benefits with a carbohydrate replacement approach, thus, setting the stage for food innovators to create palatable, convenient and healthy foods.

What are the strengths and limitations of this research?

Response:

Limitations can be found on page 10, lines 308-314. Strengths follow on line 315-319.